Precision medicine; diabetes; genetics; epigenetics; artificial intelligence

**Corresponding author:**
Xiaoying Li;
Email: li.xiaoying@zs-hospital.sh.cn

# Updates of precision medicine in type 2 diabetes

Mingfeng Xia[1] [ID] and Xiaoying Li[1,2]

[1]Ministry of Education Key Laboratory of Metabolism and Molecular Medicine, Department of Endocrinology and Metabolism, Zhongshan Hospital, Fudan University, Shanghai, China and [2]Shanghai Key Laboratory of Metabolic Remodeling and Health, Institute of Metabolism and Integrative Biology, Fudan University, Shanghai, China

## Abstract

Diabetes mellitus is prevalent worldwide and affects 1 in 10 adults. Despite the successful development of glucose-lowering drugs, such as glucagon-like peptide-1 (GLP-1) receptor agonists and sodium-glucose cotransporter-2 inhibitors recently, the proportion of patients achieving satisfactory glucose control has not risen as expected. The heterogeneity of diabetes determines that a one-size-fits-all strategy is not suitable for people with diabetes. Diabetes is undoubtedly more heterogeneous than the conventional subclassification, such as type 1, type 2, monogenic and gestational diabetes. The recent progress in genetics and epigenetics of diabetes has gradually unveiled the mechanisms underlying the heterogeneity of diabetes, and cluster analysis has shown promising results in the substratification of type 2 diabetes, which accounts for 95% of diabetic patients. More recently, the rapid development of sophisticated glucose monitoring and artificial intelligence technologies further enabled comprehensive consideration of the complex individual genetic and clinical information and might ultimately realize a precision diagnosis and treatment in diabetics.

## Impact statement

Diabetes mellitus has become a global public health crisis that affects 537 million people worldwide. Despite the great success in the development of novel glucose-lowering drugs, the proportion of patients achieving satisfactory glucose control has not risen as expected during the last decade. The heterogeneity of diabetes determines that a one-size-fits-all strategy is not suitable for people with diabetes. Our review article summarized the current progress in heterogeneity of diabetes from the perspective of genetics and epigenetics, and introduced a promising clinical substratification of type 2 diabetes. Thanks to the recent rapid development of sophisticated glucose monitoring and artificial intelligence technologies in the management of diabetes, we are able to process a large number of individual multi-dimensional genetic, anthropometric, clinical, biochemical and imaging information and make objective and correct judgment to improve the long-term outcome of diabetic patients. The emergence of new technologies might provide solutions for precision diagnosis and treatment of diabetes.

## Introduction

Diabetes mellitus is diagnosed if blood glucose concentration exceeds a threshold, which predisposes to microvascular and microvascular end-organ complications. Diabetes continues to increase in prevalence worldwide (Ingelfinger and Jarcho, 2017). Currently, diabetes affects 537 million people worldwide (IDF, 2021). Diabetes is also the leading cause of disability globally (GBD 2017 Disease and Injury Incidence and Prevalence Collaborators, 2018), as well as causing an increase in the risk of death from cardiovascular disease, renal disease, and cancer, and reducing life expectancy by 4–10 years on average (Rao Kondapally Seshasai et al., 2011). Early and intensive management of diabetes to achieve the recommended glycemic and metabolic targets can reduce long-term diabetes complications (Khunti and Millar-Jones, 2017).

In recent decades, great success has been achieved in the development of novel glucose-lowering drugs, such as glucagon-like peptide-1 (GLP-1) receptor agonists and sodium-glucose cotransporter-2 inhibitors. At present, pharmacological therapies, comprising 10 classes of medicines approved by the FDA, could be utilized to control blood glucose. However, the proportion of patients achieving satisfactory glucose control has not risen as expected (Bhat et al., 2021). The inadequate understanding of the diverse pathophysiological mechanisms and personalized treatment of diabetes partly limits our ability to treat diabetes.

The aim of our current review is to provide an overview of precision medicine in diabetes, focusing on the genetics and epigenetics, clinical stratification and personalized prevention, treatment of this disease and its related complications (Figure 1).

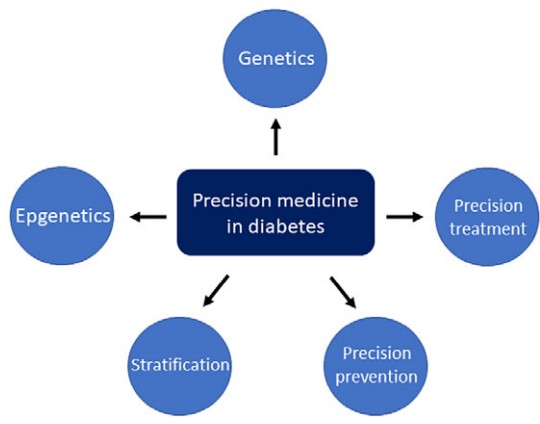

**Figure 1.** The precision medicine in diabetes management.

## Heterogeneity of diabetes

The heterogeneity of diabetes was recognized several decades ago, when diabetic patients were divided into insulin-sensitive and insulin-insensitive subgroups based on the oral glucose tolerance test (OGTT) (Himsworth and Kerr, 1939). In 1979, the classification of type 1 and type 2 diabetes was first proposed by the American Diabetes Association (National Diabetes Data Group, 1979). Meanwhile, Fajans and Tattersall described a subgroup of diabetes with inheritance across many generations called "maturity onset diabetes in the young" (MODY) (Tattersall and Fajans, 1975). However, it was not until the introduction of genomic medicine in the recent 20 years, that the molecular mechanisms of the monogenic diabetes were uncovered.

Patients with diabetes do not equally respond to glucose-lowering therapies. Thus, pharmacologic intervention should be individualized based on factors such as duration of diabetes, presence of existing comorbidities, expected duration of life, weight, age, family history of diabetes-related complications and funding for prescribed medications and technology. Previous studies indicated that approximately 30% of individuals with type 2 diabetes do not respond well to metformin, and 5% have intolerable side effects (Kahn et al., 2006; Cook et al., 2007). Clinical diabetic patients with different etiological processes, such as obesity, metabolic syndrome, beta cell dysfunction or lipodystrophy, respond differently to glucose-lowering drugs with different mechanisms (Udler and Kim, 2018). Patients from different ethnicities respond differently to glucose-lowering drugs, including dipeptidyl peptidase-4 inhibitors (Kim et al., 2013), metformin (Williams et al., 2014) and glucagon-like peptide 1 agonist (Velásquez-Mieyer et al., 2008). Thus, stratification of diabetic patients based on their genetic backgrounds and pathophysiological mechanisms could be considered for implementation of precision medicine in diabetes.

## Genetics of diabetes

Precision medication in monogenic diabetes has successfully guided clinical treatment. For example, individuals with rare mutation in *HNF1A* (MODY3), *HNF4A* (MODY1) and *ABCC8* (MODY12) are incredibly sensitive to the effects of sulfonylureas (Pearson et al., 2003). While individuals with loss-of-function mutations in the *GCK* gene are unlikely to develop diabetic complications, and have no need for unnecessary treatment (Steele et al., 2014). However, identification of monogenic diabetes still has not

solved discrepancy in the individual response to glucose-lowering medications in a large number of patients with type 2 diabetes.

In 2007, the first genome-wide association study (GWAS) in type 2 diabetes was reported (Sladek et al., 2007). There was much hope that genetics would also represent a breakthrough in understanding of the heterogeneity of type 2 diabetes at that time, however, it turned out that more than 400 genetic variants associated with type 2 diabetes to date could only explain 18% of the risk of diabetes (Mahajan et al., 2018). Moreover, every individual variant is very modestly associated with the risk of type 2 diabetes, except for the variant in the *TCF7L2* gene (Lyssenko et al., 2007). Therefore, the genomic information alone has limited value in guiding precision medicine in type 2 diabetes.

Recently, GWAS has also been performed to investigate the genetic risk alleles of type 1 diabetes (Sharp et al., 2019). Fortunately, recent studies have shown that the identified genetic variants could account for the majority of the risk of type 1 diabetes in certain populations, and type 1 diabetes genetic risk scores could well predict the risk of type 1 diabetes with both sensitivity and specificity exceeding 80% in neonatal and African-ancestry populations (Onengut-Gumuscu et al., 2019; Sharp et al., 2019). The genetics of diabetes mellitus and diabetes complications is well summarized in the literature (Cole and Florez, 2020; Riddle et al., 2020; Deutsch et al., 2022).

## Epigenetics of diabetes

Obesity and metabolic syndrome is the main risk factor for type 2 diabetes which is caused by a complex inheritance-environment interaction (Wu et al., 2014). Epigenetics represents the heritable reversible modifications to the genome associated with environmental factors and clinical phenotypes (BLUEPRINT Consortium, 2016). Epigenetics explores the mechanism in which phenotypes are changed by non-DNA sequence variation, including DNA methylation, histone modifications, non-coding RNAs regulations. Previous studies have shown that blood methylation markers in the *TXNIP*, *ABCG1*, *PHOSPHO1*, *SOCS3* and *SREBF1* genes were associated with the risk of incident type 2 diabetes (Chambers et al., 2015). Since epigenetic patterns have vast plasticity, the epigenetic alteration in type 2 diabetes could be targeted for personalized treatment. The non-coding RNAs, especially the short noncoding RNAs (microRNAs), can regulate the expression of protein-coding genes or epigenetic regulators including DNA methyltransferases, histone deacetylases and polycomb protein coding gene (Bushati and Cohen, 2007). MicroRNA changes in vivo were associated with the insulin resistance level in type 2 diabetes (Gallagher et al., 2010), and its value as biomarkers for type 2 diabetes has been investigated (de Candia et al., 2017). Integrating both genetic and epigenetic risk factors might reflect the inheritance-environment interaction, and provide a promising solution for subclassification of type 2 diabetes.

Recently, the complex association between SARS-CoV-2 Infection (COVID-19) and diabetes has emphasized the environmental factors in promoting diabetes (Shin et al., 2021; Cao et al., 2023). Studies showed that critical COVID-19 and hospitalized COVID-19 subjects had an increased risk of type 2 diabetes, and genetic liability to COVID-19 had a causal effect on type 2 diabetes (Cao et al., 2023). Mechanistically, the COVID-19 spike protein physically interacted with GRP78 protein in cell surface of adipose tissue, promoting hyperinsulinemia in adipocytes via XBP1, which may be

attributed to the development and progress of diabetes (Shin et al., 2021).

## Stratification of diabetes

Various pathogenic mechanisms and outcomes of disease have been observed in the large range of type 2 diabetes (Philipson, 2020). Thus, the stratification of type 2 diabetes may be relevant in the field of precision medicine for diabetes diagnosis. Cluster analysis based on high-dimensional data, such as electronic medical records or omics data (genomics, proteomics, metabolomics, transcriptomics, lipidomics, etc.), has been utilized to identify subtypes of type 2 diabetes (Li et al., 2015; Udler et al., 2018; Wagner et al., 2021). In 2018, a study of the Swedish population with newly diagnosed diabetes used both hierarchical and *k*-means clustering to identify five subtypes of adult-onset diabetes, named severe autoimmune diabetes (SAID), severe insulin-deficient diabetes (SIDD), severe insulin-resistant diabetes (SIRD), mild obesity-related diabetes (MOD) and mild age-related diabetes (MARD), based on six clinical variables (autoantibodies, age at diagnosis, BMI, HbA1c, C peptide together with glucose for estimation of insulin secretion, HOMA-B and insulin-sensitivity, HOMA-IS) (Ahlqvist et al., 2018). SAID was characterized by the presence of GAD autoantibodies, low insulin secretion and poor metabolic control, SIDD was characterized by low insulin secretion, poor metabolic control and increased risk of retinopathy, SIRD was characterized by severe insulin resistance, obesity, late onset and markedly increased risk of nephropathy, MOD was characterized by obesity, early onset and good metabolic control, and MARD was characterized by late onset and good metabolic control. This classification of adult-onset diabetes has been most frequently replicated in three independent cohorts from different ethnicities (Philipson, 2020). These subgroups differ in genetic predisposition to diabetes, with increased frequency of *HLA* rs2854275 variant in SAID and *TCF7L2* rs7903146 variant in SIDD, MOD and MARD (Lyssenko et al., 2007). Both SIRD and MOD patients were obese, but SIRD represented the unhealthy obesity with insulin resistance and non-alcoholic fatty liver disease and MOD represented healthy obesity without insulin resistance (Cohen et al., 2011). More importantly, the new subclassification of diabetes might guide the personalized treatment. The MOD and MARD patients usually had good metabolic control and disease prognosis, thus, it may be that these cases require less frequent glucose monitoring and could be easily managed with metformin and lifestyle intervention. Several medications with confirmed protective effects on specific vital organs, such as sodium-glucose cotransporter 2 (SGLT2) on cardiovascular and renal outcomes, might be especially suitable for SIRD (Wanner et al., 2016). Thus, the attempts on stratification of diabetes by cluster analysis might be promising for precision diagnosis of diabetes.

More recently, the application of artificial intelligence (AI) technology was able to comprehensively ingest all required parameters in supplied formats (text, image/video, biometric data) for analysis, leading to the prediction of incident diabetes (AUROCs: 0.71–0.87) (Ellahham, 2020) and risk stratification of diabetic populations (Zou et al., 2018). Recently, a study using genomic and tabular data to predict type 2 diabetes based on Recurrent Neural Networks has been reported (Srinivasu et al., 2022). The results showed that the proposed model could predict future diabetes with fair accuracy, which may be used in real-world scenarios (Srinivasu et al., 2022). The use of AI in diabetes management could

establish a more accurate and objective stratification of type 2 diabetes based on a broad range of candidate parameters.

## Precision prevention

Comprehensive management of diabetes includes diet and exercise interventions, patient education, glucose monitoring and drug treatment. By the time the diabetes is diagnosed, diabetes-related tissue damage has occurred in nearly half of the patients (Ambaby and Chamukuttan, 2008). An early intervention in patients with prediabetes, either with lifestyle interventions or pharmacologic interventions, reduces the risk of incident diabetes and improves long-term outcomes (Haw et al., 2017). However, there has been a big variation among the patients diagnosed with prediabetes in their response to lifestyle or drug intervention (Knowler et al., 2002; Knowler et al., 2009). Those who lost the least weight in the early stages of intervention showed the highest risk of incident diabetes (Delahanty et al., 2014). The reduction of body weight in patients with prediabetes after lifestyle or drug intervention was related to genetic variants (Papandonatos et al., 2015). For instance, the protective effect of metformin in reducing incidence of diabetes was associated with variation in the *SLC47A1* gene in the Diabetes Prevention Program (Jablonski et al., 2010). Therefore, the patients diagnosed with prediabetes who are unlikely to respond well to lifestyle modification might be better served by other therapeutic treatments, but more studies were required to properly identify this subgroup of prediabetes. Meanwhile, efforts have also been made to prevent the incidence of type 1 diabetes in high-risk children with at least two islet autoantibodies using dietary interventions and/or immune-targeting approaches (Skyler et al., 2018). Unfortunately, most previous intervention studies were unable to slow, halt or reverse the destruction of beta cells or delay the progression of type 1 diabetes (Hummel et al., 2011; Knip et al., 2018).

## Precision treatment

Drug treatment was recommended to achieve good glucose control and lower the risk of cardiovascular disease and specific diabetic complications in diabetic patients. Although FDA has approved 10 classes of diabetes medications, each of these medications showed great heterogeneity in therapeutic efficacy, being effective for some patients, but less effective for others, with some even experiencing adverse effects (Dennis et al., 2018). Trials of medications on diabetes have recognized that different etiologic processes of diabetes would influence the therapeutic effect of antidiabetic medications recently (Dennis et al., 2019). Reanalysis of the data from the ADOPT and RECORD studies found that the subtype of diabetic patients with insulin resistance responded better to treatment with thiazolidinediones and that older patients responded better to sulfonylureas (Dennis et al., 2019). Similar studies using prospective and primary care data in the UK found that the subgroup of diabetics with insulin resistance, obesity or high triglycerides had reduced initial response to DDP4 inhibitor and more rapid failure of therapy (Dennis et al., 2018).

There is also an ethnic difference in the individual response to a specific antidiabetic medication. Previous studies indicated that the therapeutic effect of DDP4 inhibitors is greater in Asians than in other demographic groups. Consistently, a subgroup analysis of the Trial Evaluating Cardiovascular Outcomes with Sitagliptin (TECOS) showed a greater reduction in blood glucose in East Asians

(Davis et al., 2018), and a recent REWIND study found the protective effect of Dulaglutide on cardiovascular disease is significantly stronger in the Asian Pacific area than other regions of the world (Gerstein et al., 2019). Moreover, the effect of metformin also differed among different ethnic groups, with African Americans having a greater response to metformin than was observed for European Americans (Williams et al., 2014).

The use of genetics in guiding the pharmaceutical treatment of diabetes is an important step toward the precision treatment of diabetes. A typical successful example is the application of genetics in treatment of monogenic diabetes. In that case, a specific single gene mutation is causal for the development of diabetes and targeted treatment can well bypass the etiological defect, such as the use of sulfonylurea in MODY3 caused by the mutation of *HNF1A* gene (Pearson et al., 2003). However, type 2 diabetes is much more complex than MODY, which is influenced by the complex interaction of hundreds of etiological gene variants and environmental risk factors. Traditionally, genetic studies of drug response in type 2 diabetes have focused on candidate genes known to relate to etiological processes or drug transport or metabolism. Studies have shown that a variant in the *SLC22A1* gene encoding the organic cation transporter 1 (OCT1) is involved in the cellular transport of metformin and influenced the individual response to metformin (Shu et al., 2007). Similarly, a variant in *MATE1* (Becker et al., 2009) was also associated with metformin response. Studies also found that *KCNJ11/ABCC8* risk variant increases, but *TCF7L2* risk variant reduces glycemic response to sulfonylureas (Pearson et al., 2007; Feng et al., 2008). The *PPARG* risk variant was associated with reduced glycemic response to thiazolidinediones (Kang et al., 2005). GWAS in the pharmacogenetics of diabetes made no assumptions about the drug mechanism and metabolism, and have therefore provided novel insights into genetic factors related with response to antidiabetic medication, and successfully identified variants at the *ATM* and *SLC2A2* genes as modulators of individual response to metformin (GoDARTS and UKPDS Diabetes Pharmacogenetics Study Group et al., 2011; Zhou et al., 2016). GWAS of response to other antidiabetic drugs was still necessary to establish the drug response prediction system based on genetics to guide clinical treatment.

The management of diabetes is a comprehensive approach, and glucose monitoring, patient education and lifestyle intervention also played essential roles in diabetes treatment in addition to diabetes medications. Currently, the use of remote continuous glucose monitoring (CGM) enables monitoring of a complete view of glucose control over 24 h (Battelino and Danne, 2019), and the in-depth insights of glucose and direct feedback provided by CGM system have efficaciously controlled the blood glucose and reduced the incidence of hypoglycemia (Beck et al., 2017). The accurate CGM also enabled the personalized lifestyle intervention prescription tailored to each diabetic patient. An AI-based decision support system named the Advisor Pro was recently developed. This system sends the data from CGM to a cloud server and uses AI to determine required insulin doses remotely. Studies showed that insulin doses recommended by the Advisor Pro had no significant difference compared with that given by physicians, suggesting this to be a convenient approach in managing diabetes (Nimri et al., 2018; Nimri et al., 2022). With the rapid development in the AI field, it is highly possible that AI will introduce a revolutionary shift in management of diabetes from conventional therapeutic strategies to data-driven precision treatment, based on the combination of individual genetic and glucose monitoring information and decision-making systems based on machine learning.

Lifestyle modifications and first-line medication treatment do not prevent the progressive decline of β-cell mass and function in some patients. Therefore, advanced strategies need to be developed to address this issue. Stem cell therapy for the treatment of diabetes has made great progress in recent years (Furuyama et al., 2019; Siehler et al., 2021). Both type 1 diabetes and type 2 diabetes can benefit from stem cell therapy. Studies showed that stem cell therapy increased serum C-peptide and reduced glycosylated hemoglobin (HbA1c) in subjects with type 1 diabetes or type 2 diabetes, but had no significant effect on fasting glucose (Zhang et al., 2020). Additionally, stem cell therapy improved insulin requirements in subjects with type 2 diabetes (Zhang et al., 2020). Different types of stem cells affect the clinical efficacy of therapy for diabetes. Bone marrow mononuclear cells were more effective than mesenchymal stem cells in the treatment of type 1 diabetes, whereas both bone marrow mononuclear cells and mesenchymal stem cells had favorable effects on type 2 diabetes (Zhang et al., 2020). This all suggests that stem cell therapy for the treatment of diabetes is an attractive and potential strategy, but is still facing enormous challenges, for instance, the need for greater diversity in the source of stem cells, and also inconsistency in stem cell preparation, evaluation systems and safety.

## Future outlook

The heterogeneity of diabetes creates challenges for the wider applicability of precision medicine in successful treatment of this disease. Although the implementation of precision medicine in diabetes is progressing well, more recent developments within precision medicine may benefit both newly diagnosed patients with diabetes, as well as those exposed to glycemic toxicity for years. Multidisciplinary cooperation will be conducive to further in-depth analysis and understanding of diabetes. Currently, our knowledge on the association of individual genetic background with the pathogenesis and drug response of diabetes is increasing rapidly. The application of AI in the management of diabetes enables the objective and comprehensive analysis and process of a large number of individual multi-dimensional genetic, anthropometric, clinical, biochemical and imaging information, and might provide a solution for precision diagnosis and treatment of diabetes. The concept and tools of precision medicine help to accurately predict, diagnose and treat diabetes and its complications. Although there is still a long way to go, precision medicine will become the driving force for early intervention, early prevention and accurate management of diabetes in the future.

**Open peer review.** To view the open peer review materials for this article, please visit http://doi.org/10.1017/pcm.2023.12.

**Acknowledgement.** We have obtained financial supports from National Key Research and Development Program of China (2021YFC27 00403), Shanghai Municipal Science and Technology Commission Foundation (23XD1423300).

**Competing interest.** All authors declare that there is no duality of interest.

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
