## [Reviewer Report]

*Comments to Author*: The article by Mingfeng Xia and Xiaoying Li describes that despite of the successful development of glucose-lowering drugs, such as GLP-1 receptor agonists and iSGLT-2 recently, the proportion of patients achieving satisfactory glucose control has not risen as expected. They highlight the importance of the heterogeneity of diabetes determines that a one-size-fits-all strategy. The recent progress in genetics and epigenetics of diabetes has gradually unveiled the mechanisms underlying the heterogeneity of diabetes, and cluster analysis has shown promising results in the substratification of the type 2 diabetes, the major form of diabetes. They conclude that the rapid development of the sophisticated glucose monitoring and artificial intelligence technologies further enabled comprehensive consideration of the complex individual genetic and clinical information, and might ultimately realize precision diagnosis and treatment in diabetic patients.

I have the following comments:

- The introduction should be expanded with more information and more citations. A figure in this section is necessary.

- The use of genetics in guiding the pharmaceutical treatment of diabetes is an important step towards the precision treatment of diabetes. This section has to be expanded and make a table with both in vitro and in vivo studies.

- The future outlook section has to be expanded and improved.

- The authors have to improve all sections of the article with more and more updated information.

---

## [Reviewer Report]

*Comments to Author*: Article needs to be revised completely as it is poorly written and difficult to understand.

Precision Medicine in Diabetes Review

10/29/22

Page/Line Comment

Abstract Suggest changing the term DIABETIC PATIENTS to people with diabetes

Abstract Change this sentence and diabetes is undoubtedly more

heterogeneous than the conventional subclassification, such as type 1,

type 2, monogenic and gestational diabetes.

Place a period after people with diabetes. Then edit the next sentence to: Diabetes is undoubtedly more

heterogeneous than the conventional subclassification, such as type 1,

type 2, monogenic and gestational diabetes.

Abstract. Last sentence. Change in diabetic patients to people with diabetes.

Introduction With the dramatic

escalation in obesity, diabetes has become a fast increased disease worldwide. Actually authors should consider discussing other causes progression from normal glucose regulation to diabetes other than obesity. For example, the COVID pandemic has increased the risk of developing both type 1 and type 2 patients. COVID also increases diabetes risk in YOUNGER patients. (https://www.cdc.gov/mmwr/volumes/71/wr/mm7102e2.htm)

Xie, Y. & Al-Aly, Z. Lancet Diabetes Endocrinol. https://doi.org/10.1016/S2213-8587(22)00044-4 (2022

General note Change all phrases: diabetic patients, to ‘PATIENTS WITH DIABETES.”

Introduction This phrase needs to be edited: However, if blood glucose is well controlled, the clinical outcomes of diabetic patients could be tremendously improved 5. Change to: Early and intensive management of diabetes targeting recommended glycemic and metabolic targets, can reduce long-term diabetes complications. (Khunti K, et al. Prim Care Diabetes. 2017;11(1):3-12)

Page 5 Suggested edit: Moreover, it has been well recognized that not all diabetic patients respond to the

antidiabetic treatment in the same way. Change to: Patients with diabetes do not respond to glucose lowering therapies equally. Thus, pharmacologic intervention should be individualized based on factors such as duration of diabetes, presence of existing comorbidities, expected duration of life, weight, age, family history of diabetes related complications, and ability to cover the cost of prescribed medications and technology.

6 Epigenetics of diabetes. Authors should again consider adding a statement about the effect of COVID infections on activating genes which promote the expression of diabetes.

7 This statement makes no sense. Please revise: Type 2 diabetes is an exclusion diagnosis, and about 90% of diabetes was type 2 diabetes29

7 What is this? Omics data

7 Section about stratification of diabetes.

Look, Im sorry. I read this section 5 times and was totally confused about what the authors are saying or suggesting. And, this is actually a topic that I typically understand.

9 Section precision management:

diabetic drugs should be changed to pharmacologic interventions

9 Change prediabetic patients to patients diagnosed with prediabetes.

9 Do not start a sentence with a preposition: It is not until recently that trials of

medications on diabetes recognized that different etiologic processes of diabetes would

influence the therapeutic effect of antidiabetic medications51

12 Edit this sentence: Although the implementation of precision medicine in diabetes is on the way ahead. The future application of precision medicine may benefit newly diagnosed patients with diabetes, as well as those exposed to glycemic toxicity for years.

---

## [Editor Report]

*Comments to Author*: This review has good quality, mainly for individuals with little expertise in the area or as an overview for beginners. There is a large amount of dense studies in the literature on about genomics of diabetes, and perhaps this is the main challenge for the authors. Perhaps a better delimitation of the study object could be considered to open writing space for an adequate deepening of the results obtained in each study cited in the text. Data on conclusions about each study could be more explained, sometimes it was too superficial. For example, addressing AI and genomics could have been a clipping error. If the authors propose a general review on the topic, I still believe that other relevant topics in the area were missing, as suggested below.

1. In the introduction, authors have cited that DM affects 141 million adults in China. The following sentence in the same paragraph describes “diabetes is also the ninth leading cause of population death”. Is this position in China, or globally? Please remove the numbers about your country from the text, as our journal is aimed at research groups worldwide. Your article is not exclusively about China.

2. In the section, Heterogeneity of Diabetes. The sentence: “However, it was not until the introduction of genetics in the recent 20 years, that the molecular mechanisms of the monogenic diabetes were uncovered.” Several studies on medical genetics have been performed about Mody before 2002 (for example Yagamata et al 1996). I think that the word “genetics” should be changed to “genomic medicine”.

3. In the section, Heterogeneity of Diabetes. In the last sentence, the word “crucial” should be changed to “might be considered”, since the genomic stratification of DM-patients is still a promise in personalized medicine. There is no robust recommendation for this medical approach into clinical practice to date, such as you have described in the following sections.

4. In the last sentence of the section Genetics of Diabetes you have described the value of genetic risk scores for type 1 DM. References 21 and 22 report findings in neonatal and african-descendents populations specifically. You should describe in the text that these scores were performed in specific populations, neonatal and african-ancestry populations with more details about these cohorts. Genetic risk scores are dependent on the genetic variation identified among the different populations.

5. Again, in the section Stratification of Diabetes, the word “crucial” in the text should be changed because it is very emphatic. “Thus, the stratification of type 2 diabetes is crucial in the field of precision medicine for diabetes diagnosis”. Terms such as “may be relevant”, or “may be considered” is preferably in this incipient medical context.

6. There is no reference supporting the sentence “The MOD and MARD patients usually had good metabolic control and disease prognosis, thus required less frequent glucose monitoring and could be easily managed with metformin and lifestyle intervention”. Please cite the reference(s) or adjust the text, for example; “thus, in these cases it might require less frequent…” and cite your reference, or “thus, in our opinion, in these cases it might require less frequent…” although this last suggestion maybe would be inappropriate.

7. In the last paragraph of the section Stratification of Diabetes, is there some article about AI and diabetes taking into consideration genomic variants? If yes, you should cite these articles, describing if the results were consistent or not. If there is no publication on it, please report that there is no evidence to date.

8. Several continuous glucose monitoring devices have been approved by FDA, not only Advisor Pro as cited in the last section of your revision. Further, you have not cited any trial or study showing a particular relevance of this particular device against the others. How was the reason to cite this specific device? You can cite that several devices have been approved by FDA and that in general the system sends the GCM data for a cloud server…

9. I missed some reference in your article regarding stem cell therapy and diabetes. Several recent studies have demonstrated consistent clinical efficacy in controlling diabetes, including in children. I believe that in your review there may be something about this. You mentioned AI as an additional theme in your revision about genomic medicine, so I believe that stem cells is a topic related to precision medicine that deserves some explanation in your revision.

---

## [Reviewer Report]

*Comments to Author*: This is a poorly written manuscript which requires editing. I am not going to spend time editing this paper. I do believe, based on the first 7 pages I reviewed, that the manuscript is not fit for publication. I spent an hour looking at the first 5 pages then gave up.

Precision Medicine Manuscript Review # 2

Jeff Unger, MD

Page number/Line Comment

3/3 This line is poorly written. Just remove this line and the reference: Diabetes has become a fast increased disease worldwide1

5/4 This sentence needs to be edited: The heterogeneity of diabetes has been recognized several decades ago, which divided people with diabetes into insulin-sensitive and insulin-insensitive subgroups based on the oral glucose tolerance test (OGTT):

Several decades ago, the heterogeneity of glucose intolerance was recognized as a disease state dividing individuals into insulin-sensitive and insulin- insensitive subgroups based on one’s oral glucose tolerance test (OGTT). 7

5/20 This paragraph, again, is poorly written and needs to be edited. I am NOT going to edit this paper.

However, it was not until the introduction of genomic 10 medicine in the recent 20 years, that the molecular mechanisms of the monogenic diabetes were uncovered.

6/12 Change “might be considered” to SHOULD be considered

6/18 Change super sensitive to “highly sensitive”

6/19 Another poorly written sentence:

While individuals with loss-of19 function mutations in the GCK gene are unlikely to develop diabetic complications, and have no need for unnecessary treatment

Suggest removing the word: WHILE

7/20 Poorly written: In 2007, the first genome-wide association study (GWAS) in type 2 diabetes was reported

Should say: The first genome-wide association study (GWAS) in type 2 diabetes was published in…

---

## [Editor Report]

*Comments to Author*: Dear colleagues,

We would like to congratulate the dedicated effort on this manuscript. However, we believe that the article does not fit within the proposal and scientific rigor of our journal.

The article is not accepted for publication and we wish you success in future submissions.

Sincerely.